# Preoperative Hilar and Mediastinal Lymph Node Staging in Patients with Suspected or Diagnosed Lung Cancer: Accuracy of 18F-FDG-PET/CT:A Retrospective Cohort Study of 138 Patients

**DOI:** 10.3390/diagnostics13030403

**Published:** 2023-01-22

**Authors:** Fuad Damirov, Karen Büsing, Gökce Yavuz, Rudolf Hatz, Farkhad Manapov, Julia Michels, Peter Hohenberger, Eric Roessner

**Affiliations:** 1Department of Thoracic Surgery, Ludwig Maximilian University of Munich, 81377 Munich, Germany; 2Department of Surgery, Division of Surgical Oncology and Thoracic Surgery, University Hospital Mannheim, University of Heidelberg, 68167 Mannheim, Germany; 3Clinic for Radiology and Nuclear Medicine, University Hospital Mannheim, University of Heidelberg, 68167 Mannheim, Germany; 4Department of Radiation Oncology, Ludwig Maximilian University of Munich, 81377 Munich, Germany; 5Department of Pulmonology, University Hospital Mannheim, University of Heidelberg, 68167 Mannheim, Germany; 6Department of Pulmonology, Thoraxklinik Heidelberg, University of Heidelberg, 69126 Heidelberg, Germany; 7Department of Thoracic Surgery, Center for Thoracic Diseases, University Medical Center of the Johannes Gutenberg University Mainz, 55131 Mainz, Germany

**Keywords:** lung cancer, staging, 18F-FDG-PET/CT scan, diagnostic accuracy, lymph nodes

## Abstract

The aim of this study was to evaluate the diagnostic accuracy of integrated 18F-fluorodeoxyglucose positron emission computed tomography (18F-FDG-PET/CT) in hilar and mediastinal lymph node (HMLN) staging of suspected or proven lung cancer, and to investigate potential risk factors for false negative and false positive HMLN metastases. We retrospectively analyzed 162 consecutive patients with suspected or pathologically proven non-small cell lung cancer (NSCLC). The receiver operating characteristic (ROC) curve was generated to determine the diagnostic efficacy of 18F-FDG-PET/CT. Univariate and multivariate analyses were conducted to detect risk factors of false positives and false negatives. The sensitivity, specificity, positive predictive value (PPV), negative predictive value (NPV), and accuracy of integrated 18F-FDG-PET/CT in detecting HMLN metastases were 59.1% (26/44), 69.1% (65/94), 47.3% (26/55), 78.3% (65/83), and 65.9% (91/138), respectively. The ROC curve showed an area under the curve (AUC) of 0.625 (95%-CI 0.468–0.782). The incidence of false negative and false positive HMLN metastases was 21.7% (18/83) and 52.7% (29/55), respectively. Our data shows that integrated 18F-FDG-PET/CT staging provides lower specificity and sensitivity. This confirms the ESTS guideline on lymph node staging for PET-positive HMLN. Yet it advocates more invasive staging even for PET-negative HMLN.

## 1. Introduction

Lung cancer is the most common form of malignancy and remains the leading cause of death among all cancers. Proper staging of patients with lung cancer would increase the chances of survival. Thus, accurate evaluation of metastases in HMLN is a decisive diagnostic point, which may dictate operability and survival in patients with NSCLC. Metastasis to mediastinal lymph nodes is mostly an inoperable condition [1,2]. Neoadjuvant chemotherapy with surgical resection or simultaneous chemoradiotherapy are reliable solutions for patients with metastatic lymph nodes [3]. Therefore, proper staging of NSCLC adds critical prognostic value and chooses the best therapy concept [4]. CT with contrast is widely adopted for the staging of NSCLC, however, does not have adequate reliability to identify lymph node metastases with sufficient accuracy [5]. Because of the use of size criteria, the diagnostic efficiency of CT in detecting mediastinal lymph node metastases is poor [6]. Magnetic resonance imaging (MRI) does not increase diagnostic efficiency over CT [7]. Different from CT, 18F-fluorodeoxyglucose positron emission tomography (18F-FDG-PET) is a functional imaging method that is based on the enhanced glucose metabolism of malignant tissue and cells [8]. To minimize the essential limitations of 18F-FDG-PET, such as deficient quality of morphological imaging report, other imaging modalities applying integrated 18F-FDG-PET/CT were engineered [9]. After the development of integrated 18F-FDG-PET/CT, anatomical imaging reports can be linked to TNM staging, thereby an integrated 18F-FDG-PET/CT was implemented into the guidelines [10]. Even though several earlier studies have shown that the integrated 18F-FDG-PET/CT is more helpful for detecting HMLN metastasis, outcomes regarding the extent of advantages of integrated 18F-FDG-PET/CT are conflicting [11]. Moreover, the occurrence of occult lymph node metastasis in NSCLC patients presenting subtle uptake by integrated 18F-FDG-PET/CT is 7–16% [12,13,14,15], and false positive reports from inflammatory lesions are still uncertain. The main goal of this study is to investigate the accuracy of integrated 18F-FDG-PET/CT in the detection of HMLN metastases in patients with suspected NSCLC and to investigate potential risk factors for false negative and false positive findings. Our secondary goal is to confirm the role of invasive staging in proving integrated 18F-FDG-PET/CT reports.

## 2. Materials and Methods

### 2.1. Patient Eligibility

A review was undertaken for patients with confirmed or suspected NSCLC who underwent surgery from January 2012 to December 2017 at the division of Surgical Oncology and Thoracic Surgery of University Hospital Mannheim. All patients who received neoadjuvant chemotherapy or radiotherapy and patients without intraoperative lymph node sampling and patients without integrated 18F-FDG-PET/CT scanning were excluded. The remaining 138 consecutive patients with suspected or proven NSCLC underwent staging with integrated 18F-FDG-PET/CT prior to lung resection. The diagram illustrating patient enrollment at study entry is shown in Figure 1. Twelve of seventeen patients with suspected clinical PET N2 situation (cN2) and all patients with suspected clinical PET N3 situation (cN3) were down staged after endobronchial ultrasound needle biopsy or mediastinoscopy prior to lung resection. All patients underwent systematic lymph node dissection during surgery. Disease stage was evaluated, according to the TNM Classification of Malignant Tumors, 8th Edition [16]. 

Patient medical documentation was evaluated for the following information: sex (male or female), age, smoking status (never, ex- and current smoker), history of lung disease or coexisting diabetes, tumor laterality (right or left side), lobar distribution of tumor, histological type and grade, tumor size (cm), visceral pleural invasion (VPI) (presence or absence), lymph node metastasis (pN0, pN1, or pN2) and maximum standardized uptake value (SUVmax) of primary tumor. The details of the patient’s characteristics and tumor evaluations are shown in Table 1. Histological classification of NSCLC is based on the World Health Organization (WHO) classification [17]. Preoperative and postoperative staging are based on the TNM staging system [18,19,20]. The local Ethical Committee approval was not required owing to the retrospective, observational, and anonymous nature of this study. The Patient Consent Statements were not obtained due to anonymous nature of the study. 

### 2.2. Integrated 18F-FDG PET/CT

An integrated 18F-FDG-PET/CT was performed with a Biograph Molecular CT system (Biograph mCT Siemens), consisting of a PET detector with 4 rings, 48 detector blocks in each ring and an integrated 64-slice CT for attenuation correction of PET data. Patients were asked to fast for 6 h prior to PET/CT scan and only glucose-free water was allowed. An intravenous injection of 5–7 MBq kg^−1^ of 18F-FDG/kg of body weight was administered, and patients rested for approximately 60 min before scanning. PET/CT acquisitions of patients were performed in supine position with their arms raised above the head; whole body images (from the proximal femur to the head) were acquired. PET and CT images were reconstructed and interpreted by a qualified nuclear physician for each case. The maximum standardized uptake value (SUVmax) was measured with a region-of-interest (ROI) technique and calculated by the software according to standard formulas. Pulmonary, hilar, and mediastinal LN stations were deemed positive for metastatic spread if they exhibited focally increased FDG uptake higher than the normal background activity, as determined by qualitative analysis. Median time interval between 18F-FDG-PET/CT and surgery was 36 days. An example image of cN1 and cN2 situation on integrated 18F-FDG-PET/CT scan is shown in Figure 2.

### 2.3. Surgical Resection and Histopathology

All patients underwent anatomical lung resection and radical lymphadenectomy at the division of Surgical Oncology and Thoracic Surgery of University Hospital Mannheim. Systematic lymph node dissection or sampling was carried out in stations 2R, 4R, 3A, 3P, 7–13 in right-sided tumors, and in 4L, 5, 6, 7–13 for left-sided tumors if available. All resected tumor specimens (primary tumor characteristics and lymph node status) were examined by experienced pulmonary pathologists in standard techniques, and immunohistochemistry was used when appropriate. Histological classification of NSCLC was based on the WHO classification.

### 2.4. Data Analysis

Diagnostic characteristics of the integrated 18F-FDG-PET/CT were assessed on a per-patient basis. The sensitivity, specificity, PPV, NPV, and their 95% confidence intervals (CIs) and accuracy of integrated 18F-FDG-PET/CT in the assessment of intrathoracic lymph node involvement were calculated in accordance with the following formulae:Sensitivity = TP/(TP + FN) ± 1.96√ ((Se(1 − Se))/(TP + FN)) 
Specificity = TN/(TN + FP) ± 1.96√ ((Sp(1 − Sp))/(TN + FP)) 
Positive predictive value = TP/(TP + FP) ± 1.96√ ((PPV(1 − PPV))/(VP + FP)) 
Negative predictive value = TN/(TN + FN) ± 1.96√ ((NPV(1 − NPV))/(TN + FN)) 
where TP = true positive, TN = true negative, FP = false positive, FN = false negative, Se = sensitivity, Sp = specificity, PPV = positive predictive value, and NPV = negative predictive value. 

Continuous variables are presented as median values and quartiles. Dichotomous variables were analyzed by contingency tables using the Chi^2^-Test statistics. The findings were compared between groups using the Mann–Whitney U-test. Multivariate analysis was performed by binary logistic regression analysis. To assess optimal cut-off values of SUVmax of LN, receiver–operator characteristics (ROC) analysis and the Youden criterion were used. Statistical analysis was performed using SPSS Version 26 (IBM, Armonk, NY, USA). Results with type I error *p* < 0.05 were considered significant.

## 3. Results

### 3.1. Patient Characteristics

All demographic, operative, and histopathological data of the enrolled 138 patients (56 women, 82 men; mean age 64 years, range 29–84) are summarized in Table 1. The median tumor size was 2.8 cm, and the median SUVmax of the primary tumor was 8.1. Of these 138 patients, 69 (50%) had adenocarcinoma, 48 (34.8%) had squamous cell carcinoma, 14 (10%) had other types of malignancies, and seven (5%) had no malignancies.

### 3.2. Evaluation of HMLN Status by Pathological Examination and Integrated 18F-FDG-PET/CT

In pathologic staging, 94 (68%) patients had no lymph node involvement (Table 2). All evaluating indicators of integrated 18F-FDG-PET/CT examination were analyzed in Table 3. The sensitivity, specificity, PPV, NPV, and accuracy of integrated 18F-FDG-PET/CT in detecting HMLN metastases were 59.1% (26/44), 69.1% (65/94), 47.3% (26/55), 78.3% (65/83), and 65.9% (91/138), respectively. The ROC curve based on the SUVmax of HMLN status is shown in Figure 3. The SUVmax of nodules had an area under the curve (AUC) of 0.625 (95%-CI 0.468–0.782) with a cut-off SUVmax of 4.7. If nodal uptake with SUVmax > 4.7 was interpreted as positive the sensitivity, specificity, PPV, NPV, and accuracy of integrated 18F-FDG-PET/CT were 50.0%, 81.5%, 70.6, 64.7%, and 66.6% (34/51).

### 3.3. Risk Factors Associated with False Negative Detection of HMLN Metastasis of 18F-FDG-PET/CT Staging

The incidence of occult HMLN metastasis in this study was 21.7% (18/83). Of 18 patients with lymph node metastasis, multi-station metastasis was found in 4 patients, while the other 14 patients showed metastasis on a single station. Appendix A summarizes the preoperative 18F-FDG-PET/CT scanning and the patterns of HMLN involvement. Thirteen of eighteen patients had metastasis in the hilar lymph node. Three patients had concurrent diabetes although they had normal levels of fasting blood glucose before the PET/CT scan. Four patients had concurrent lung disease. All 18 patients did not show any pathological FDG uptake in HMLN in integrated 18F-FDG-PET/CT.

Table 4 summarizes the results of univariate analysis for factors associated with HMLN metastasis in 83 18F-FDG-PET/CT negative patients. Factors significantly associated with HMLN metastasis were tumor size (>3.0 cm, *p* = 0.008), presence of VPI (*p* = 0.020), and SUVmax of tumor (≥8.25, *p* = 0.026). The multivariate analysis (Table 5) identified the SUVmax of the tumor (≥8.25, *p* = 0.032) and the presence of VPI (*p* = 0.010) as risk factors for false negative HMLN metastasis.

### 3.4. Risk Factors Associated with False Positive Detection of HMLN Metastasis of 18F-FDG-PET/CT Staging

A total of 55 patients were diagnosed as positive cases by integrated 18F-FDG-PET/CT. Postoperatively, 52.7% (29/55) of patients were confirmed as false positives. Appendix A summarizes the preoperative 18F-FDG-PET/CT scanning of all false positive patients and the patterns of HMLN involvement. The univariate and multivariate analysis identified the SUVmax of LN (<4.7, *p* = 0.011) as the only risk factor for false positive HMLN metastasis. Some results of these analyses are displayed in Table 6 and Table 7.

## 4. Discussion

An important part of the initial staging of patients with lung cancer is the determination of HMLN metastasis. If more than one lymph node station is involved (N2 or N3), neoadjuvant treatment consisting of chemo-, chemoimmunotherapy, and/or chemoradiotherapy should be considered in the multidisciplinary tumor board. In recent years, integrated 18F-FDG-PET/CT is standard hybrid imaging for the initial staging of NSCLC. In a meta-analysis from Zhao et al., the pooled sensitivity and specificity of integrated 18F-FDG-PET/CT staging on a per-patient basis were 71.9% and 89.9%, respectively [11]. This suggested that integrated 18F-FDG-PET/CT had more specificity but less sensitivity for LN staging than CT scans. Thus, we conducted the retrospective study to further analyze the sensitivity and specificity of integrated 18F-FDG-PET/CT of HMLN staging. The false negative and false positive findings were 21.7% (18/83) and 52.7% (29/55), respectively. Integrated 18F-FDG-PET/CT provides low sensitivity (59.1%), specificity (69.1%), and accuracy (65.9%) for HMLN staging of NSCLC.

In order to maximize the accuracy of integrated 18F-FDG-PET/CT, ROC curve analysis was performed. With an AUC of 0.625, the variables in our study are poorly suited for determining an optional cut-off value. Nevertheless, the curve identified a low optional cut-off SUVmax value of HMLN of 4.7, which maximized specificity. When a SUVmax of 4.7 or greater is used to classify any LN as positive and less than 4.7 as negative on integrated 18F-FDG-PET/CT, the specificity of all patients is 81.5%.

In the present study, occult LN metastases were found in 18 of 83 patients (21.7%); thus, NPV was 78.3%. Five of eighteen patients had pN2 disease, and two of five patients had more than one pN2 LN station involvement. This finding was concordant with a study by Gomez-Caro et al. that reported a 32% prevalence of false negative diagnoses [21]. In a study by Li et al. the frequency of false negative nodal findings in patients with NSCLC was 13.2% [15]. 

In the recent study, the presence of VPI and the SUVmax of the primary tumor were independent predictors of false negative LN metastasis for patients with NSCLC by integrated 18F-FDG-PET/CT in both univariate and multivariate analysis. 

The risk factor of VPI in occult lymph node metastasis may support the assumption by Shimizu et al. that there is a possible VPI tumor cell pathway through the subpleural lymphatics into the mediastinal lymph nodes [22]. 

It is similar to previous studies [14,15], that a high SUVmax (≥8.25) of the primary tumor was also determined as a significant risk factor for occult LN metastasis in our study, although wide heterogeneity is observed in the definition of an SUV threshold among studies [9,23].

In the current study, up to 52.7% of patients (29/55) were upstaged from cN0 to cN1 or cN2 and cN3 by integrated 18F-FDG-PET/CT scan. The accuracy of integrated 18F-FDG-PET/CT staging in the detection of true positive LNs interferes with difficulties due to the increased glycolytic activity of inflammatory tissue and benign tumors, in addition to that of malignant tumors. The false positive upstaging of LNs may have been caused by active inflammation due to granulomatous diseases or reactive hyperplasia [24]. The only risk factor for false positive findings in integrated 18F-FDG-PET/CT was SUVmax of LN (<4.7). Therefore, we strongly recommend biopsies of all suspicious LNs including those with a high SUVmax that should not be equated with malignancy until tissue confirmation is obtained.

Interesting results regarding the diagnostic impact of EBUS-TBNA in addition to integrated 18F-FDG-PET/CT in stage III NSCLC were also reported by Guberina et al. Study evaluated 675 LN stations (291 positives for tumor cells) with EBUS-TBNA and integrated 18F-FDG-PET/CT and found a significant increase in the false discovery rate (5.3 to 69.1%) of PET/CT depending on anatomical mapping of the LN to echelon 1 (ipsilateral hilus), echelon 2 (ipsilateral central mediastinum) or echelon 3 (upper ipsilateral mediastinum, contralateral mediastinum, and hilus). The study concluded that the false discovery rate of integrated 18F-FDG-PET/CT increases continuously with distance from the primary tumor [25].

The main limitation of the current study is the retrospective nature of the work. It is necessary to provide prospective or randomized trials to clarify the true incidence of a false negative and false positive diagnosis of LN metastases of NSCLC patients scanned by integrated 18F-FDG-PET/CT. Another limitation of this study is related to the statistical power of our results, particularly for ROC curve analysis. It was caused by the absence of SUVmax of LNs in radiological reports of integrated 18F-FDG-PET/CT scanning if subtle uptake appeared. 

## 5. Conclusions

The sensitivity, specificity, and accuracy of integrated 18F-FDG-PET/CT for HMLN staging in patients with operable NSCLC are low. In the presence of positive HMLNs, invasive mediastinal LN staging must be performed to exclude a possible false positive upstaging of integrated 18F-FDG-PET/CT. This confirms the ESTS guideline on lymph node staging for PET-positive HMLN. Nevertheless, it supports more invasive staging even for PET-negative HMLNs.

## Figures and Tables

**Figure 1 diagnostics-13-00403-f001:**
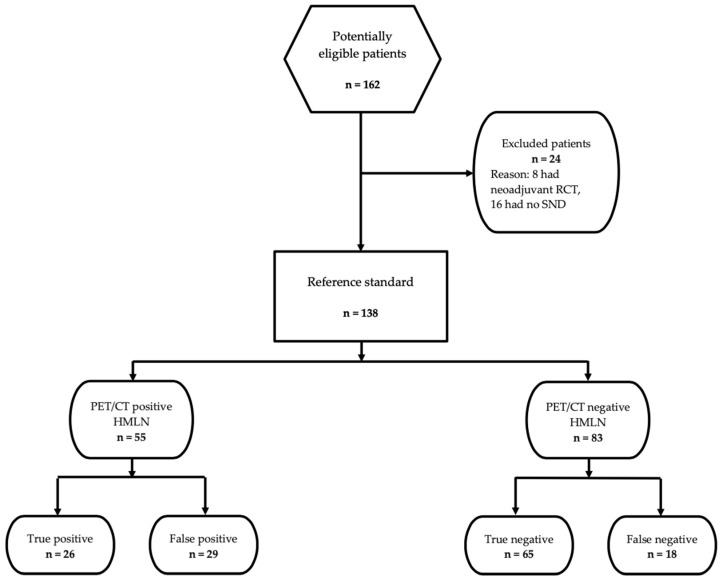
STARD flow diagram illustrating patient enrollment at study entry. PET: positron emission tomography; SND: systematic lymph node dissection; RCT: radiochemotherapy.

**Figure 2 diagnostics-13-00403-f002:**
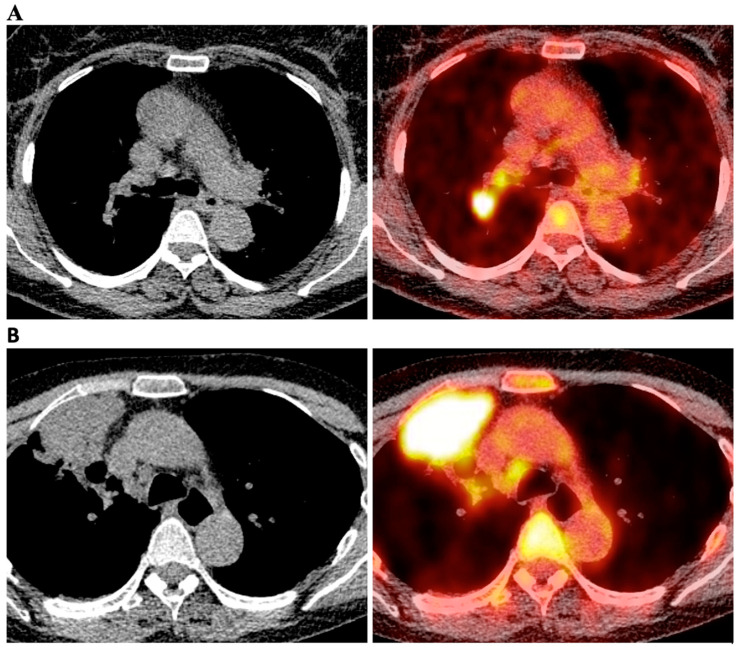
Integrated 18F-FDG-PET/CT scans of the chest. **A**—clinical N1 situation. **B**—clinical N2 situation.

**Figure 3 diagnostics-13-00403-f003:**
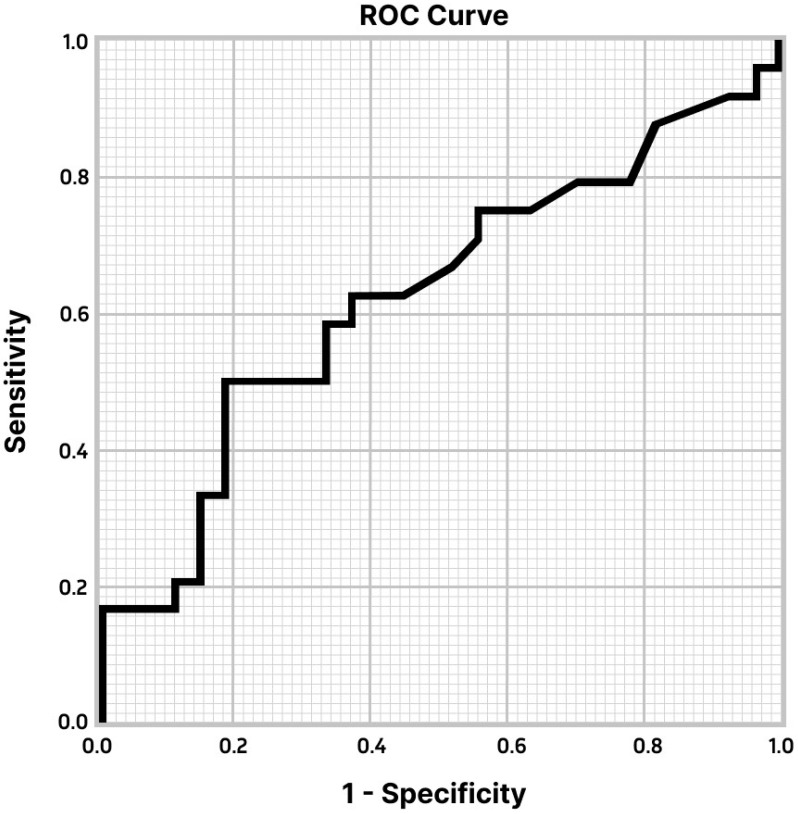
ROC curve for SUVmax of HMLN. The curve had an AUC of 0.625 (95% CI 0.468–0.782) with a cut-off SUVmax 4.7.

**Table 1 diagnostics-13-00403-t001:** Characteristics of patients and tumors (*n* = 138). RUL: right upper lobe, RML: right middle lobe, RLL: right lower lobe, LUL: left upper lobe, LLL: left lower lobe, SD: standard deviation.

Characteristics	Distribution (n)
Sex	
Male/Female	82/56
Age	
Mean ± SD/Range	64 ± 9.25/29 − 84
Smoking status	
Smoker/former smoker/never smoker	72/48/18
History of lung disease or diabetes	
Yes/no	67/71
Location of the tumor	
Central/non-central	33/105
Lobar distribution of the tumor	
RUL/RML/RLL/LUL/LLL	40/8/30/41/19
Histological type	
Adenocarcinoma/squamous cell carcinoma/other types/no malignancy	69/48/14/7
Pleural invasion	
Yes/No	20/118
Pathological T stage	
Tis/T1a/T1b/T1c/T2a/T2b/T3/T4	1/12/18/30/24/13/24/9
Lymph node metastasis	
pN0/pN1/pN2	94/27/17
Pathological stage	
0/Ia1/Ia2/Ia3/Ib/IIa/IIb/IIIa/IIIb/IVa	1/5/16/18/17/6/34/25/3/6
Tumor size (cm)	
Median/Mean ± SD/Range	2.8/3.164 ± 1.823/0.3 − 12.5
SUVmax of primary tumor	
Median/Mean ± SD/Range	8.1/9.623 ± 6.342/1.1 − 34.77

**Table 2 diagnostics-13-00403-t002:** Correlation between clinical PET N factor and pathological N factor.

	pN0	pN1	pN2
cN0 (*n* = 83)	65	13	5
cN1 (*n* = 25)	9	12	4
cN2 (*n* = 17)	9	1	7
cN3 (*n* = 13)	11	1	1

**Table 3 diagnostics-13-00403-t003:** Histological findings and integrated 18F-FDG-PET/CT results of HMLN. TP: true positive, FN: false negative, FP: false positive, TN: true negative.

	Pathological Positive (N1 + N2)	Pathological Negative (N0)
PET/CT (+)	26 (TP)	29 (FP)
PET/CT (−)	18 (FN)	65 (TN)

**Table 4 diagnostics-13-00403-t004:** Univariate analysis for factors associated with false negatives in integrated 18F-FDG-PET/CT negative patients.

Variable	Status	Pathologically Positive	Pathologically Negative	*p*-Value
Age (years)	>65	10	33	n.s.
	≤65	8	32	
Gender	Female	7	26	n.s.
	Male	11	39	
Smoking status	Never-smoker	2	12	n.s.
	Ex-smoker	7	21	
	Current smoker	9	32	n.s.
Concurrent lung disease	Absent	14	45	
	Present	4	20	n.s.
Concurrent diabetes	Absent	15	53	
	Present	3	12	n.s.
Tumor laterality	Right	9	34	
	Left	9	31	n.s.
Tumor location	Central	4	11	
	Non-central	14	54	n.s.
Other malignancy history	Absent	8	35	
	Present	10	30	n.s.
Tumor size	≤3.0 cm	6	44	
	>3.0 cm	12	21	0.008
VPI	Absent	12	58	
	Present	6	7	0.020
SUVmax of primary tumor	<8.25	6	39	
	≥8.25	12	23	0.026
Grade	well	0	12	
	moderate/poor	18	50	0.059

**Table 5 diagnostics-13-00403-t005:** Multivariate analysis for factors associated with false negatives in integrated 18F-FDG-PET/CT negative patients.

Variable	Odds Ratio	Confidence Interval	*p*-Value
SUVmax of primary tumor (≥8.25)	0.170	0.034–0.857	0.032
VPI	0.109	0.20–0.586	0.010

**Table 6 diagnostics-13-00403-t006:** Univariate analysis for factors associated with false positives in integrated 18F-FDG-PET/CT positive patients.

Variable	Status	Pathologically Positive	Pathologically Negative	*p*-Value
Age (years)	>65	12	7	0.086
	≤65	14	22	
Gender	Female	12	11	n.s.
	Male	14	18	
Smoking status	Never-smoker	2	2	n.s.
	Ex-smoker	9	11	
	Current smoker	15	16	n.s.
Concurrent lung disease	Absent	16	15	
	Present	10	14	n.s.
Concurrent diabetes	Absent	24	27	
	Present	2	2	n.s.
Tumor laterality	Right	16	18	
	Left	10	11	n.s.
Tumor location	Central	10	8	
	Non-central	16	21	n.s.
Other malignancy history	Absent	17	20	
	Present	9	9	n.s.
Tumor size	≤3.0 cm	15	14	
	>3.0 cm	11	14	n.s.
VPI	Absent	23	25	
	Present	3	4	n.s.
SUVmax of primary tumor	<8.25	10	13	
	≥8.25	15	16	n.s.
Grade	well	2	8	
	moderate/poor	24	19	0.076
SUVmax of LN	≥4.7	12	5	
	<4.7	12	22	0.017

**Table 7 diagnostics-13-00403-t007:** Multivariate analysis for factors associated with false positives in integrated 18F-FDG-PET/CT positive patients.

Variable	Odds Ratio	Confidence Interval	*p*-Value
SUVmax of LN (<4.7)	0.061	0.007–0.531	0.011

## Data Availability

The dataset used and/or analyzed during the current study are available from the corresponding author on reasonable request.

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
