# Peer review of "Preoperative Hilar and Mediastinal Lymph Node Staging in Patients with Suspected or Diagnosed Lung Cancer: Accuracy of 18F-FDG-PET/CT:A Retrospective Cohort Study of 138 Patients"

_diagnostics, 2023, doi:10.3390/diagnostics13030403_

Round 1
Reviewer 1 Report
The manuscript deals with the evaluation of diagnostic accuracy for 18F-FDG-PET/CT in hilar and mediastinal lymph node (HMLN) staging. A patient cohort of 180 was analyzed for various factors such as false positives, false negatives, sensitivity, specificity, etc. The study indicates that 18F-FDG-PET/CT provides less sensitivity and specificity, and more invasive methods need to be considered. Overall, the highlights of the study are relevant and well-described. The manuscript may be accepted with the following major revisions.
1. The grammar and sentence formation need to be improved and the manuscript should undergo thorough proofreading with professionals to improve the overall quality of the manuscript. For example:
Line 56: “Even though outcomes regarding the extent of advantages of integrated 18F-FDG-PET/CT are conflicting” is inconclusive.
Line 60: “The main goal of this study was to investigate the accuracy of integrated…” The tense should be “is”. Same with line 63.
2. Abbreviations should be included when it is included for the first time. For example: Lines 74 and 75: cN2 and cN3 (clinical PET N factor)
3. Include reference for lines 77-78.
4. Table 6: Some numbers were underlined for tumor laterality and tumor location for "pathologically negative" column. Please provide justification for underlining these numbers.
5. The manuscript mentions histopathology for resected tumor specimens. A representative image for such studies should be provided.
6. In materials and methods, ethical considerations should be included. Does this study have IRB approval? Also, please mention whether patient consent forms were obtained prior to the study.
7. The manuscript is based on integrated 18F-FDG-PET/CT. Please provide a representative image for the scan.
8. The abstract last sentence and the conclusion section's last sentence are the same. “But it advocates more invasive staging even for PET negative HMLNs.” Please change the wording to get distinct sentences.
Reviewer 2 Report
Fuad Damirov et al wrote a manuscript on the performance of PET in correctly identifying LN metastasis in patients with suspected or confirmed lung cancer. Although it has been known that PET has its false positive and negative performance this paper should be praised for identifying the exact PPV & NPV, and their attempt to identify the risk factors associated with false positivity and negativity.
I have the following comment:
1. There are minor typos that needs to be corrected (e.g., not small cell lung cancer in abstract)
2. I believe that, since the paper is about PET & CT, the sample size cannot be taken as 180, but at most 162 according to the STARD flowchart.
3. May I just confirm that ALL 138 patients that eventually underwent analysis have PROVEN lung cancer in the primary site (i.e., "Other types of 21 in Table 1 refers to other type so CANCER histology)?
4. It would be interesting to know what the alternate diagnosis were in those false positive cases. Please include if you have the data.
5. May I know if the sample size
6. On that note, the regional prevalence of TB and sarcoidosis will affect the SEN/SPC/PPV/NPV. This should be included in the sample size calculation.
Reviewer 3 Report
This is my review on “Preoperative hilar and mediastinal lymph node staging in patients with suspected or diagnosed lung cancer: Accuracy of 18F-FDG-PET/CT. A retrospective cohort study of 138 patients”. The diagnostic accuracy of integrated 18F-fluorodeoxyglucose positron emission computed tomography was evaluated on the case of hilar and mediastinal lymph node for lung cancer. The 180 patients that this report includes were suspected or proven cases of not small cell lung cancer.
Introduction is fine and the aim of the study is clearly declared. Authors also tried to confirm the role of invasive staging in proving integrated 18F-FDG-PET/CT reports.
Materials and methods are adequately described. The flow diagram is well presented and table 1 presents all the necessary information.
Results are interesting and with easy-to-read tables. The ROC curve should be improved in terms of pixel-quality.
Discussion is based on the results and is fine. Authors presented clearly their limitations.
Round 2
Reviewer 1 Report
The major revisions have been incorporated into the manuscript.